# Carotid artery stenosis and ischemic cerebrovascular events after radiotherapy in patients with head and neck cancer

**Nawaphan Taengsakul**[1]*, **Padungcharn Nivatpumin**[1], **Thong Chotchutipan**[2], **Sunanta Tungfung**[2]

**1** Department of Surgery, Chulabhorn Hospital, Princess Srisavangavadhana College of Medicine, Chulabhorn Royal Academy, Bangkok, Thailand, **2** Department of Radiation Oncology, Chulabhorn Hospital, Chulabhorn Royal Academy, Bangkok, Thailand

* nawaphan.tan@cra.ac.th

**Data Availability Statement:** All relevant data are within the manuscript and its Supporting Information files.

## Abstract

Radiotherapy is the main treatment for patients with head and neck cancer (HNC) and is associated with an increased risk of ischemic cerebrovascular events (ICVE). The purpose of this cross-sectional study was to determine the incidence of ICVE and carotid artery stenosis (CAS) in patients with HNC who receive radiotherapy and the risk factors for CAS. We enrolled 907 patients with HNC who underwent radiotherapy between February 2011 and June 2022 and obtained information on their clinical and tumor characteristics and their treatment from the clinical records. Data on risk factors for atherosclerosis, medications used, and radiotherapy were also collected. The patients were followed through to the end of 2023 unless they died or were lost to follow-up. The overall incidence of ICVE was 1.98%, with a cumulative incidence of 1.65% over 5 years. In patients who did not have a preexisting carotid artery lesion, the cumulative incidence of significant CAS was 1.3% at 12 months, 2.2% at 24 months, and 2.5% at 36 months post-radiotherapy. The most important risk factors for new CAS were age >65 years (aHR = 2.60, p = 0.008, 95% confidence Interval: 1.28–5.30), laryngeal cancer (aHR = 2.36, p<0.017, 95% confidence Interval: 1.01–5.55), and total plaque score (aHR = 1.38, p<0.001, 95% confidence Interval: 1.23–1.56). There was a significant increase in stenosis, plaque score, and wall thickness in all areas in the carotid artery (p<0.001). The incidence of ICVE and the cumulative incidence of CAS was found to be lower in the Thai population than in other populations. The main risk factors for new CAS were age >65 years, laryngeal cancer, and total plaque score. Changes in the carotid artery were detected early and affected all areas in the artery. Patients with HNC treated by radiotherapy should be assessed for risk factors for CAS and undergo vascular surveillance during follow-up.

## Introduction

Radiotherapy is recommended in approximately 80% of patients with head and neck cancer (HNC) and may be combined with surgery or systemic treatment [1,2]. Survival rates are

**Funding:** The author(s) received no specific funding for this work.

**Competing interests:** The authors have declared that no competing interests exist.

gradually improving as a result of more intensive regimens, and there is an increasing emphasis on preventing the long-term adverse events of radiotherapy in these patients [3]. The salivary glands and structures involved in swallowing are particularly vulnerable to radiotherapy-related adverse events, the most common of which are xerostomia, thickened saliva, and dysphagia [2,4]. Radiotherapy has also been associated with an increased risk of ischemic cerebrovascular events (ICVE), including ischemic stroke and transient ischemic attacks [5]. Radiation impairs blood vessels via various mechanisms, including direct damage that causes intimal hyperplasia, necrosis of the media layer, and fibrosis [6] and indirect damage to the primary vessel by destruction of the vasa vasorum [7]. Studies in patients who have undergone radiotherapy for HNC have shown an increased incidence of carotid artery stenosis (CAS) that ranges from 12% to 30% [8–12]. One study found that 38% of patients with HNC who developed CAS after radiotherapy had more than 50% stenosis [13], which may reflect the higher radiotherapy doses used nowadays and increasing patient survival. ICVE are twice as likely to occur after radiotherapy in the head and neck area [14,15] and are considered clinically relevant in view of their devastating impact on quality of life [16]. The majority of ICVE affect the anterior circulation, which is supplied by the carotid arteries, and there is consensus that radiotherapy to the head and neck increases the risk of ICVE [17–19]. The purpose of this study was to determine the incidence of ICVE and of CAS in patients who receive radiotherapy for HNC and the risk factors for CAS.

## Materials and methods

### Study design and population

The study had a retrospective cohort design and included 907 patients with HNC involving the oral cavity, pharynx, larynx, nasal cavity, or salivary glands. The missing data from incomplete follow-up was 44 cases (4.8%) due to a desire to follow up at their nearby hospital that was missing. The study was approved by the Chulabhorn Institutional Ethics Committee for Human Research (approval number 048/2565). The study was registered with Thai Clinical Trial Registry Number (TCTR) 20230914002 and Clinicaltrial.gov Identifier: NCT06556979. All patients underwent radiotherapy to the head and neck region between February 2011 and June 2022 with or without surgical resection and were investigated by computed tomography (CT) before and after radiotherapy as Fig 1. The data was accessed for research purpose on 31 October 2022. Patients with lymphoma and those with previous treatment for CAS or radiotherapy for other diseases were excluded. We obtained information on patient background factors, tumor characteristics, and treatment from the medical records. Risk factors for atherosclerosis, including hypertension, dyslipidemia, diabetes mellitus, coronary artery disease, peripheral arterial disease, and cigarette smoking, were identified. Baseline laboratory data, including hemoglobin $A_{1C}$, fasting blood sugar, lipid profile, serum creatinine level, and estimated glomerular filtration rate, were obtained, as was information on treatment, including antiplatelet agents, statins, angiotensin-converting enzyme inhibitors (ACEIs), calcium channel blockers (CCBs), and anticoagulants. The type and stage of HNC, indication for radiotherapy, type of radiotherapy, total cumulative dose, and number of courses were also recorded. Unfortunately, the laboratory data were incomplete in many cases, so could not be subjected to statistical analysis. Follow-up was continued until loss to follow-up, death, or the end of 2023, whichever came first. Confidentiality of the data was secured by assigning of a code for each patient record and all data were fully anonymized before accessing process. Images were analyzed to determine the timing of development of CAS, identify significant (>50%) CAS using the North American Symptomatic Carotid Endarterectomy Trial criteria, and calculate the total plaque score (TPS) and wall thickness. The carotid vessels were divided into the

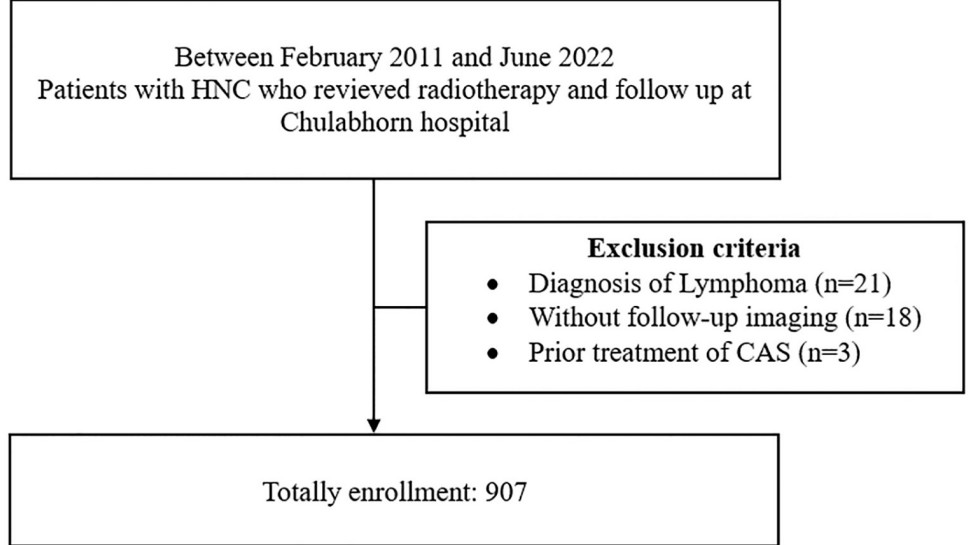

**Fig 1. Patients' enrollment HNC, head and neck cancer; CAS, coronary artery stenosis.**

common carotid artery (CCA), carotid bulb, internal carotid artery (ICA), and external carotid artery (ECA). The bulb was defined as the portion of the artery 1 cm caudally to 1 cm cranially from the point where the CCA divided into the ICA and ECA. The right and left carotid artery systems were each divided into the following five segments: proximal CCA ($\geq$20 mm proximal to bulb), distal CCA (<20 mm proximal to bulb), carotid bulb, ICA. and ECA. Each segment was graded as follows: 0, normal or no detectable plaque; 1, plaque occupying <30% of vessel diameter; 2, plaque occupying 30%–49% of vessel diameter; 3, plaque occupying 50%–69% of vessel diameter; 4, plaque occupying 70%–99% of vessel diameter; and 5, 100% occlusion of vessel diameter by plaque. The carotid plaque score was obtained for each patient by summing the scores obtained for the five arterial segments in both carotid arteries. Wall thickness and degree of CAS were measured in the five segments of the carotid artery at baseline and during follow-up. Differences in these values were calculated. CT was performed at least twice. The initial CT scan was performed within 2 months before initiation of radiotherapy and follow-up CT was performed 1 month after completing radiotherapy. In cases with multiple follow-up CT scans, the last scan was used to evaluate post-radiation change. The 1-year follow-up CT scan was also examined. All baseline and follow-up measurements were made by a vascular surgeon who was blinded to the clinical data.

## Measurement of outcomes

The primary endpoint was the incidence of ICVE (ischemic stroke or transient ischemic attack in the anterior circulation) following completion of radiotherapy. The secondary endpoint was the cumulative incidence of CAS at 12, 24, and 36 months after radiotherapy and risk factors that were identified to influence development of new CAS.

## Statistical analysis

The statistical analysis was primarily descriptive. Quantitative variables are presented as the mean ± standard deviation and categorical variables as the number (percentage). We defined preexisting CAS and risk factors for atherosclerosis, and only extracted data for patients who

did not have carotid artery lesions before radiotherapy to determine the risk of developing CAS. Continuous variables were compared between groups using the independent *t*-test and Mann–Whitney *U* test. The chi-squared test was used to examine the relationship between outcome variables. Fisher's exact test was used when the data were limited. Differences in treatment outcomes were calculated with 95% confidence intervals (95% CIs). We utilized the multivariable Cox proportional hazard regression to get the adjusted hazard ratio and survival analysis after defining our data to be right censored. Atherosclerosis risk factor and cancer type did not interact, according to our evaluation of risk factor interactions to identify confounding factors by the multivariate Cox regression model. Overall survival was defined as the interval between the day of initiation of radiotherapy and the day of the last follow-up. Cumulative incidence was examined using the Kaplan–Meier method. All statistical analyses were performed using STATA version 16.0 (StataCorp, College Station, TX, USA). A two-sided P-value of <0.05 was considered statistically significant.

## Results

The patient and tumor characteristics are shown in Table 1. Approximately three-quarters of the patients were male and under the age of 65 years. More than half of the patients were of normal weight, with one-quarter being underweight. The three most common comorbidities were hypertension, diabetes mellitus, and dyslipidemia. Approximately 10% of the study population were on CCB, ACEI, antiplatelet, or statin therapy. Sixty percent had pharyngeal cancer, 20.84% had cancer of the oral cavity, and 11.47% had laryngeal cancer. More than half of the patients had TNM stage 4 disease and approximately 20% had TNM stage 3. One-third of patients had poorly differentiated squamous cell carcinoma. Eighteen percent had preexisting CAS, which was significant in 3.63%. Following radiotherapy, the CAS rate increased from 18% to 23%. Table 1 also provides information on oncological treatment, imaging findings regarding the carotid artery lesions, treatment of these lesions, ischemic cerebrovascular events, and causes of death.

Table 2 shows the demographic and clinical characteristics of the patients in whom CAS was identified before radiotherapy. The prevalence of pre-existing CAS was 18.7%, which was significant in 3.6% of cases. More than half of the patients had a normal weight (body mass index 18.5–24.9). Nearly 75% were male and aged ≥65 years. One-quarter of the patients were underweight, 15% were overweight, and 4% were obese. Most did not have underlying disease. The most prevalent comorbidities were hypertension, diabetes, and dyslipidemia. Patients who were older (particularly >65 years) and those with a higher TPS (≥7) were significantly more likely to have preexisting CAS or significant (>50%) CAS. Hypertension and diabetes mellitus were associated with a higher prevalence of carotid artery lesions, including any CAS and significant CAS, on imaging before radiotherapy. The prevalence of dyslipidemia and peripheral arterial disease was only significantly higher if CAS was significant. Any CAS and significant CAS were less likely in patients whose current medications included an ACEI and a statin than in those who were not taking these agents. However, rates of any CAS and significant CAS were higher in patients taking an antiplatelet agent or a CCB. CAS was significantly less common in patients with pharyngeal cancer, but significant (>50%) CAS was not. Patients with laryngeal or salivary gland cancer had significantly higher rates of CAS, but only laryngeal cancer was associated with a significantly higher rate of significant CAS. Patients with oral or nasal cavity cancer tended to have lower rates of CAS, but this finding was not significant. There was no significant difference in the stage of cancer between the group with no CAS, the group with any CAS, and the group with significant (>50%) CAS. The mean TPS was 0.90 ± 2.49 mm, with 96.14% having a score <7 and 8.86% having a score ≥7. The TPS was

**Table 1. Baseline characteristics of the study population.**

| Variable | | All patients (n = 907) |
|---|---|---|
| **Sex** | Male | 653 (72.00) |
| | Female | 254 (28.00) |
| **Age** | All patients | 56.81 ± 14.42 |
| | ≤65 years | 655 (72.22) |
| | >65 years | 252 (27.78) |
| **Body mass index** | All patients | 21.61 ± 5.31 |
| | Underweight (<18.5) | 225 (25.77) |
| | Normal weight (18.5–24.9) | 475 (54.41) |
| | Overweight (25.0–29.9) | 134 (15.35) |
| | Obesity (>30) | 39 (4.47) |
| **Comorbidities** | Diabetes mellitus | 99 (10.92) |
| | Hypertension | 207 (22.82) |
| | Dyslipidemia | 57 (6.28) |
| | Coronary artery disease | 1 (0.11) |
| | Peripheral arterial disease | 18 (1.98) |
| | Valvular heart disease | 19 (2.09) |
| | Thyroid disease | 16 (1.76) |
| **Smoking** | | 536 (59.10) |
| **Current medication** | Antiplatelet agent | 100 (11.03) |
| | Anticoagulant | 12 (1.32) |
| | ACEI | 125 (13.78) |
| | CCB | 137 (15.10) |
| | Statin | 97 (10.69) |
| **Tumor location** | Oral cavity | 189 (20.84) |
| | Pharynx | 547 (60.31) |
| | Larynx | 104 (11.47) |
| | Nasal cavity | 40 (4.41) |
| | Salivary gland | 34 (3.75) |
| **TNM stage** | In situ | 1 (0.11) |
| | 1 | 83 (9.18) |
| | 2 | 118 (13.05) |
| | 3 | 186 (20.58) |
| | 4 | 516 (57.08) |
| **T stage** | 0 | 2 (0.22) |
| | 1 | 214 (23.67) |
| | 2 | 236 (26.11) |
| | 3 | 192 (21.24) |
| | 4 | 260 (28.76) |
| **N stage** | 0 | 281 (31.08) |
| | 1 | 168 (18.58) |
| | 2 | 340 (37.61) |
| | 3 | 115 (12.72) |
| **M stage** | 0 | 867 (95.91) |
| | 1 | 37 (4.09) |
| **Pathological report** | Well differentiated SCC | 178 (19.63) |
| | Moderately differentiated SCC | 214 (23.59) |
| | Poorly differentiated SCC | 295 (32.52) |
| | SCC with differentiation not defined | 171 (18.85) |

(*Continued*)

**Table 1.** (Continued)

| Variable | | All patients (n = 907) |
|---|---|---|
| **Treatment for cancer** | Surgery | 167 (22.42) |
| | Chemotherapy | 568 (76.24) |
| | Cisplatin | 461 (61.88) |
| | Carboplatin | 176 (23.62) |
| | 5-FU | 176 (23.62) |
| | Paclitaxel | 25 (3.36) |
| | Gemcitabine | 22 (2.95) |
| **Radiotherapy** | Aim | |
| | Definitive | 576 (77.32) |
| | Adjuvant | 134 (17.99) |
| | Palliative | 46 (6.17) |
| | Number of radiation courses | |
| | 1 | 872 (96.11) |
| | 2 | 35 (3.89) |
| | Radiation technique | |
| | IMRT | 52 (6.98) |
| | 3D-CRT | 50 (6.71) |
| | VMAT | 643 (86.31) |
| | Total radiation dose, Gy | 69.30 ± 24.29 |
| **Pre-radiation imaging findings** | CAS | 170 (18.74) |
| | Significant CAS | 33 (3.63) |
| | Total plaque score | 0.90 ± 2.49 |
| | <7 | 872 (96.14) |
| | ≥7 | 35 (8.86) |
| | Wall thickness (mm) | 1.11 ± 1.37 |
| **Post-radiation imaging findings** | New carotid lesion radiation | 179 (23.43) |
| | Progression of carotid lesion | 153 (20.05) |
| **Treatment of carotid lesions** | Carotid artery stenting | 2 (0.22) |
| | Best medical treatment | 177 (20.69) |
| **ICVE** | All | 18 (1.98%) |
| | Transient ischemic attack | 1 (6.67) |
| | Ischemic stroke | 17 (94.44) |
| | Associated with carotid artery lesion | 1 (6.67) |
| **Survival and mortality** | Median overall survival, months | 161 |
| | Overall mortality | 279 (30.76) |
| | ICVE-related | 7 (0.77) |
| | Cancer-related | 272 (29.98) |

The data are presented as the mean ± standard deviation or as the number (percentage) unless otherwise stated. 3D-CRT, three-dimensional conformal radiation therapy; 5-FU, 5-fluorouracil; ACEI, angiotensin-converting enzyme inhibitor; CAS, coronary artery stenosis; CCB, calcium channel blocker; ICVE, ischemic cerebrovascular events; IMRT, intensity-modulated radiation therapy; SCC, squamous cell carcinoma; VMAT, volumetric-modulated arc therapy; Gy, Gray; mm, milimeter.

0.08 ± 0.66 in the group with no CAS, 4.47 ± 3.93 in the group with any CAS, and 8.39 ± 4.60 in the group with significant CAS.

Of the 737 patients who did not have a carotid lesion before radiotherapy, 609 were regularly followed up at our hospital and 128 at an alternative hospital. Sixty-two patients

**Table 2. Demographic, clinical, and tumor characteristics in the groups without CAS, any CAS, and significant CAS before radiotherapy.**

| Variable | | All (n = 907) | No CAS (n = 737) | Any CAS (n = 170) | p-value | Significant CAS (n = 33) | p-value |
|---|---|---|---|---|---|---|---|
| Sex | Male | 653 (72.00) | 525 (71.23) | 128 (75.29) | 0.288 | 25 (75.76) | 0.616 |
| | Female | 254 (28.00) | 212 (28.77) | 42 (24.71) | | 8 (24.24) | |
| Age, years | | 56.81 ± 14.42 | 53.94 ± 13.56 | 69.64 ± 10.10 | <0.001 | 74.09 ± 8.63 | <0.001 |
| | ≤65 | 655 (72.22) | 598 (81.14) | 57 (33.33) | <0.001 | 4 (12.12) | <0.001 |
| | >65 | 252 (27.78) | 139 (18.86) | 113 (66.47) | | 29 (87.88) | |
| Body mass index | | 21.61 ± 5.31 | 21.74 ± 5.11 | 20.68 ± 5.03 | 0.018 | 20.31 (17–24) | 0.675 |
| Comorbidity | Diabetes mellitus | 99 (10.92) | 69 (9.36) | 30 (17.65) | 0.002 | 9 (27.27) | 0.002 |
| | Hypertension | 207 (22.82) | 137 (18.59) | 70 (41.18) | <0.001 | 21 (63.64) | <0.001 |
| | Dyslipidemia | 57 (6.28) | 44 (5.97) | 13 (7.65) | 0.417 | 6 (18.18) | 0.014 |
| | CAD | 1 (0.11) | 0 (0.00) | 1 (0.59) | 0.187 | 1 (3.03) | 0.037 |
| | PAD | 18 (1.98) | 11 (1.49) | 7 (4.12) | 0.059 | 4 (12.12) | 0.003 |
| | Valvular heart disease | 19 (2.09) | 15 (2.04) | 4 (2.35) | 0.768 | 1 (3.03) | 0.491 |
| Smoking | | 536 (59.10) | 419 (56.85) | 117 (68.82) | 0.004 | 22 (66.67) | 0.366 |
| Medication | Antiplatelet agent | 100 (11.03) | 63 (8.55) | 37 (21.76) | <0.001 | 13 (39.39) | <0.001 |
| | Anticoagulant | 12 (1.32) | 7 (0.95) | 5 (2.94) | 0.056 | 2 (6.06) | 0.068 |
| | ACEI | 125 (13.78) | 655 (88.87) | 127 (74.71) | <0.001 | 13 (39.39) | <0.001 |
| | CCB | 137 (15.10) | 95 (87.11) | 128 (75.29) | <0.001 | 14 (42.42) | <0.001 |
| | Statin | 97 (10.69) | 672 (91.18) | 138 (81.18) | <0.001 | 10 (30.30) | 0.001 |
| Tumor location | Oral cavity | 189 (20.84) | 146 (19.81) | 43 (25.29) | 0.117 | 7 (21.21) | 0.939 |
| | Pharynx | 547 (60.31) | 470 (63.77) | 77 (45.29) | <0.001 | 15 (45.45) | 0.074 |
| | Larynx | 104 (11.47) | 69 (9.36) | 35 (20.59) | <0.001 | 8 (24.24) | 0.044 |
| | Nasal cavity | 40 (4.41) | 36 (4.88) | 4 (2.35) | 0.211 | 1 (3.03) | 1.000 |
| | Salivary gland | 34 (3.75) | 23 (3.12) | 11 (6.47) | 0.038 | 2 (6.06) | 0.355 |
| TPS | | 0.90 ± 2.49 | 0.08 ± 0.66 | 4.47 ± 3.93 | <0.001 | 8.39 ± 4.60 | <0.001 |
| | <7 | 872 (96.14) | 735 (99.73) | 137 (80.59) | <0.001 | 17 (51.52) | <0.001 |
| | ≥7 | 35 (8.86) | 2 (0.27) | 33 (19.41) | | 16 (48.48) | |

The data are presented as the mean ± standard deviation or as the number (percentage) unless otherwise stated. ACEI, angiotensin-converting enzyme inhibitor; CAD, coronary artery disease; CAS, coronary artery stenosis; CCB, calcium channel blocker; PAD, peripheral arterial disease; TPS, total plaque score.

(approximately 10%) had newly diagnosed CAS post-radiotherapy, with only three having significant (>50%) stenosis. Age>65 years, laryngeal cancer, and higher total plaque score were associated with an increased risk of developing a carotid artery lesion but the treatment with cisplatin was reduced risk (Table 3).

CAS was significantly more common in the proximal CCA, carotid bulb, and ICA (Table 4).

The overall incidence of ICVE was 1.98% as Fig 2, with a 5-year cumulative incidence of 1.65%. The patients with ICVE had a mean age of 63.23 years (range 58–88). Two-thirds of these patients were smokers, and 88% had multiple comorbidities, particularly hypertension and diabetes. Before onset of ICVE, half of the patients were treated with aspirin and statins for comorbidities. Seventeen patients were diagnosed with ischemic stroke and one with a transient ischemic event. All patients with ICVE received medication. Only six patients with ICVE had CAS, with four having <50% stenosis and two having newly developed stenosis. CAS progressed in four patients (from preexisting stenosis to significant stenosis in two cases and developed de novo in two cases). One of the two patients who developed significant stenosis went on to have clinical ICVE and the other remained asymptomatic.

**Table 3. Risk factors for development of CAS in patients with no carotid artery lesions before radiotherapy.**

| Variable | | All patients (n = 609) | No CAS after radiotherapy (n = 547) | CAS after radiotherapy (n = 62) | Adjusted Hazard ratio (95%CI) | p-value |
|---|---|---|---|---|---|---|
| **Age, years** | | 53.66 ± 13.06 | 52.85 ± 13.13 | 60.82 ± 9.92 | 1.08 (1.06, 1.10) | <0.001 |
| | ≤65 | 503 (82.59) | 461 (84.28) | 42 (67.74) | 1 | |
| | >65 | 106 (17.41) | 86 (15.72) | 20 (32.26) | 2.60 (1.28, 5.30) | 0.008 |
| **Comorbidity** | Diabetes mellitus | 58 (9.52) | 47 (8.59) | 11 (17.74) | 1.57 (0.66, 3.69) | 0.304 |
| **Medication** | Antiplatelet agent | 51 (8.37) | 41 (7.50) | 10 (16.13) | 1.19 (0.48, 2.95) | 0.712 |
| **Tumor location** | Larynx | 57 (9.36) | 45 (8.23) | 12 (19.35) | 2.36 (1.01, 5.55) | 0.049 |
| **Stage** | 1 | 51 (8.39) | 41 (7.51) | 10 (16.13) | 1 | |
| | 2 | 82 (13.46) | 78 (14.26) | 4 (6.45) | 0.60 (0.16, 2.26) | 0.454 |
| | 3 | 134 (22.04) | 123 (22.53) | 11 (17.74) | 0.93 (0.33, 2.65) | 0.894 |
| | 4 | 341 (56.09) | 304 (55.68) | 37 (59.68) | 1.28 (0.48, 3.42) | 0.616 |
| **Total radiation dose** | | 69.90 ± 11.50 | 69.62 ± 10.66 | 72.31 ± 17.13 | 0.99 (0.97, 1.01) | 0.310 |
| **Chemotherapy** | Cisplatin | 398 (65.35) | 365 (66.73) | 33 (53.23) | 0.43 (0.22, 0.86) | 0.016 |
| **Total plaque score** | | 0.05 ± 0.51 | 0.03 ± 0.46 | 0.21 ± 0.81 | 1.38 (1.23, 1.56) | <0.001 |

The data are presented as the mean ± standard deviation or as the number (percentage) unless otherwise stated. CAS, coronary artery stenosis.

During a follow-up of 2.10 ± 2.36 years, carotid artery lesions progressed in 16.86% of cases, leading to significant stenosis in 3.7% as Fig 3. The cumulative incidence of significant (>50%) CAS was 1.3% at 12 months, 2.2% at 24 months, and 2.5% at 36 months. Only one patient developed symptomatic CAS, despite having significant stenosis before radiotherapy, and was treated by carotid artery stenting. The stenosis was found in both the bulb and the ICA, with stenosis in the bulb increasing from 41% to 84% and stenosis in the ICA decreasing by 76% from that seen on pre-radiotherapy imaging. This patient was diagnosed with stage 2 tongue cancer and received definitive chemoradiotherapy consisting of volumetric-modulated arc therapy (VMAT; 70 Gy [2.12 Gy in 33 fractions]) and cisplatin. Thirty-three cases of

**Table 4. Frequency of carotid artery lesions identified before radiotherapy and at one month and one year after radiotherapy.**

| Finding of a carotid artery lesion | Before radiotherapy | One month after radiotherapy | One year after radiotherapy | p-value |
|---|---|---|---|---|
| **Location of CAS** | | | | |
| Proximal CCA | 12 (1.63) | 19 (2.59) | 31 (4.22) | <0.001 |
| Distal CCA | 24 (3.27) | 22 (2.99) | 31 (4.22) | 0.203 |
| Carotid bulb | 125 (17.01) | 151 (20.54) | 185 (25.17) | <0.001 |
| ICA | 48 (6.53) | 56 (7.62) | 73 (9.93) | <0.001 |
| Total plaque score | 0.39 (0.00–10.00) | 0.49 (0.00–13.00) | 0.67 (0.00–13.00) | <0.001 |
| **TPS** | | | | |
| <7 | 710 (96.60) | 700 (95.24) | 687 (93.47) | <0.001 |
| ≥7 | 25 (3.40) | 35 (4.76) | 48 (6.53) | |
| **Mean wall thickness (range)** | 0.72 (0.00–67.00) | 0.88 (0.25–2.10) | 0.98 (0.26–2.34) | <0.001 |
| Proximal CCA | 0.57 (0.17–3.43) | 0.92 (0.20–3.14) | 1.01 (0.20–3.14) | <0.001 |
| Distal CCA | 0.57 (0.15–1.93) | 0.92 (0.20–3.64) | 1.02 (0.20–3.80) | <0.001 |
| Carotid bulb | 0.59 (0.19–2.70) | 0.97 (0.20–2.97) | 1.10 (0.25–2.97) | <0.001 |
| ICA | 0.47 (0.14–2.46) | 0.77 (0.22–2.88) | 0.85 (0.00–2.80) | <0.001 |

The data are presented as the mean ± standard deviation or as the number (percentage) unless otherwise stated. CCA, common carotid artery; ICA, internal carotid artery; TPS, total plaque score.

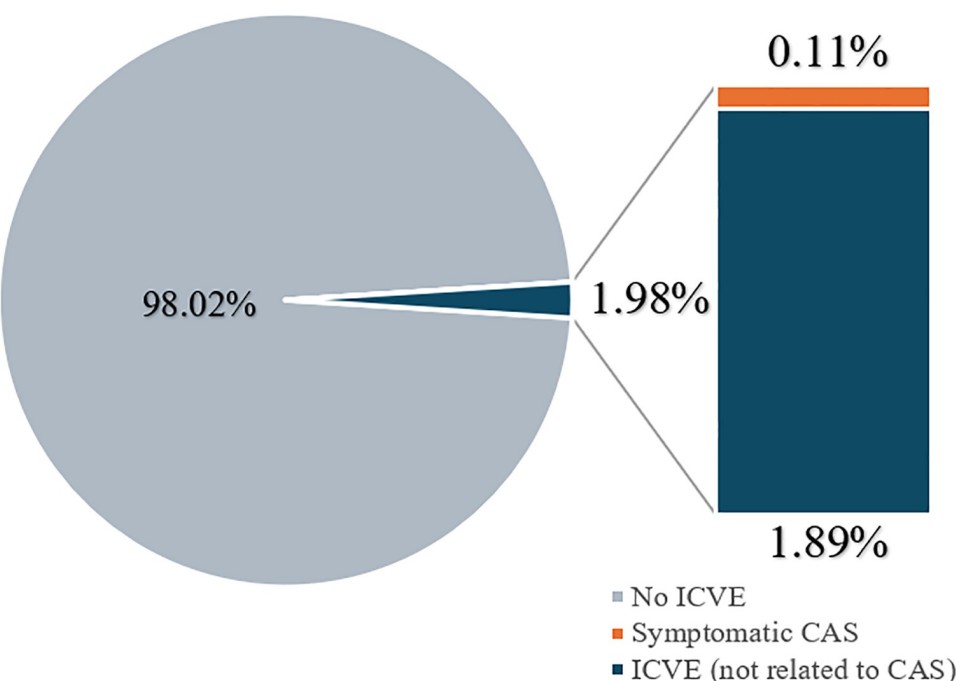

**Fig 2. Relationship between incidence of ischemic cerebrovascular events and carotid artery stenosis.** ICVE, ischemic cerebrovascular events; CAS, carotid artery stenosis.

significant but asymptomatic CAS were identified, with the majority receiving best medical treatment. There were 10 cancer-related deaths. Only two of these patients were treated by carotid artery stenting.

## Discussion

In this study, we found that the incidence of ICVE after radiotherapy for HNC was only 1.98% during a mean follow-up of 3.34 ± 3.14 years, with only one case (0.11%) of symptomatic CAS after treatment. A previous literature review found that radiotherapy to the head and neck area increased the risk of neurological events by at least two-fold [5]. Furthermore, the risk of stroke was reported to be higher in patients who have received radiation treatment than in the general population, with the risk increasing with time since completion of radiotherapy [20–22]. We summarized the previous study and our study as shown in Table 5. We found that the large retrospectives in the USA showed a greater incidence of prevalence for only stroke [22], however, this study omitted information on follow-up duration and radiation dosage Two publications, including Van Aken et al. [22] and Makita C. et al. [23], had a similar level of percentage that involved stroke and transient ischemic attack (TIA). We discovered that Van Aken et al. [22] had a lower radiation dose, which may have contributed to their slightly lower incidence of ICVE than Makita C. et al. [23] publication. Tan TH. et al.'s prior study with a 10-year incidence of stroke and no definition of radiation dose had the lowest incidence of ICVE [24]. Despite a higher radiation exposure, as seen in Table 5, our investigation revealed the lowest 5-year cumulative incidence of both stroke and TIA when compared to the prior study.

Dorresteijn et al. investigated the risk of ischemic stroke in patients under the age of 60 years who had received radiotherapy for HNC and reported a significant relative risk (RR) of 5.6 [25]. In their study, the median interval between treatment and onset of stroke was 11

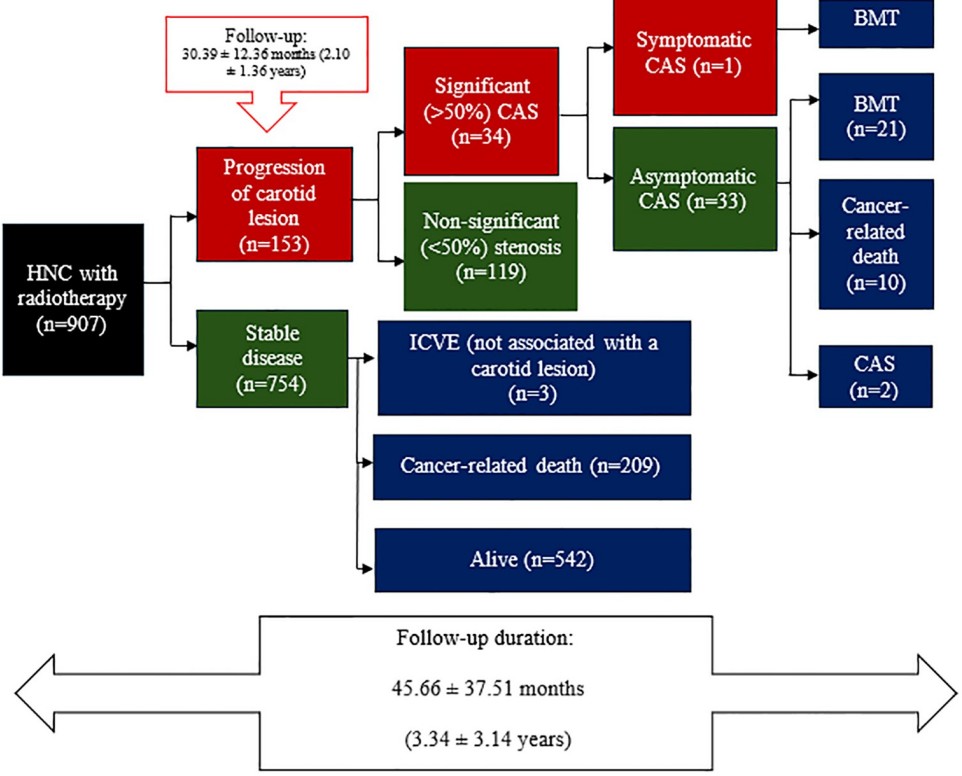

**Fig 3. Diagram showing progression to carotid artery stenosis after radiotherapy, including significant and non-significant lesions, clinical presentation, and treatment.** BMT, best medical treatment; CAS, coronary artery stenosis; HNC, head and neck cancer; ICVE, ischemic cerebrovascular events.

**Table 5. Summary of studies regarding cumulative incidence of ischemic cerebrovascular event (ICVE).**

| Study | Study design | Country | Radiation dose | Imaging | Follow-up time | Incidence |
|---|---|---|---|---|---|---|
| Ischemic cerebrovascular event (ICVE) | | | | | | |
| Our study | Retrospective cohort study (n = 907) | Thailand | 69.30 ± 24.29 Gy | CT | 3.34 ± 3.14 years | 5-year cumulative incidence of ICVE = 1.65% |
| Sun L et al. (2023) [21] | Retrospective cohort study (n = 35,897) | USA (82% White individuals, Asian16.6% Black 0.6%) | - | - | - | Cumulative incidence of stroke • 5-year = 7.37% • 10-year = 12.52% |
| Van Aken et al. (2021) [22] | Retrospective analysis of prospective cohort (n = 750) | Netherland | 39.8 ± 0.5 Gy | - | 3.4 years (0.1–10.6 years) | Cumulative incidence of ICVE • 5-year = 4.6% • 8-year = 7.4% |
| Makita C et al. (2020) [23] | Retrospective cohort study (n = 111) | Japan | 66 Gy (range, 60–74) | - | 60 months | The vascular event occurrence rate was 5.4% within 5 years and 10.7% within 8 years. |
| Tan TH et al. (2020) [24] | Retrospective cohort study (n = 3,849) | Singapore | - | - | 48.4 months (19.8–92.9 months) | 10-year cumulative incidence of stroke of 5.7% |

The data are presented as the mean ± standard deviation or as the number (percentage) unless otherwise stated. ICVE, Ischemic cerebrovascular event; Gy, Gray; CT, computer tomography.

years, with an exponential increase in risk after 10 years. In contrast, several studies have reported a lower median time to development of first stroke of 4–10 years [23–26]. One of these reports was the registry study in Singapore, which found an age-standardized incidence rate ratio of 2.54 for development of stroke in patients with NPC in comparison with the general population [24].

The main causes of stroke after radiotherapy to the head and neck are CAS, radiation vasculitis, and carotid dissection [27]. A meta-analysis by Liao et al. found that the risk of CAS in patients with NPC was four times higher after radiotherapy [28]. Another meta-analysis of 22 studies reported the prevalence of significant (>50%) CAS to be 26% in patients with HNC treated by radiotherapy and the risk of radiotherapy-related vasculopathy (CAS >50%) to be 7-fold higher than in the general population and in patients with HNC who received treatment other than radiotherapy [29]. On the other hand, recent studies revealed that radiation is not accelerating the progression of CAS in patients who are at risk for atherosclerosis [30] similar to our results. Another meta-analysis reported respective 1-year, 2-year, and 3-year incidence rates of 4%, 12%, and 21% [31], suggesting that the risk of CAS increases with time elapsed since completion of radiotherapy [32]. However, our study found lower cumulative incidence rates of significant CAS after radiotherapy (1.3% at 12 months, 2.2% at 24 months, and 2.5% at 36 months). Table 6 shows CAS was reported to have a prevalence of CAS (>50%) at a comparable rate. The cumulative incidence for more extended follow-ups as 5 years, 8 years, and 10

**Table 6. Summary of studies regarding prevalence and cumulative incidence of carotid artery stenosis (CAS).**

| Study | Study design | Country | Radiation dose | Imaging | Follow-up time | Incidence/ Prevalence |
|---|---|---|---|---|---|---|
| Carotid artery stenosis (CAS) | | | | | | |
| Our study | Retrospective cohort study (n = 907) | Thailand | 69.30 ± 24.29 Gy | CT scan | 3.34 ± 3.14 years | Cumulative incidence of significant (>50%) CAS <br> • 12 months = 1.3% <br> • 24 months = 2.2% <br> • 36 months = 2.5% |
| Carpenter et al. (2023) [33] | Retrospective (n = 628) | USA | - | DUS 94%, CTA4%, MRA1% | 4.8 years | Cumulative incidence of asymptomatic CAS (>50%) <br> • 5 years = 17% <br> • 10 years = 29.6% |
| Liu et al. (2022) [29] | Meta-analysis (n = 35,160) | USA, Hongkong, Taiwan, Italy, Canada, Malaysia, China, Germany | Vary (from >35 to >60 Gy) | Mostly DUS Only 1 study: MRA | - | Prevalence for carotid stenosis (>50%) = 26% |
| Texakalidis P et al. (2020) [31] | Meta-analysis (n = 1479) | Germany, Canada, USA, China, Taiwan, Malaysia, Italy, France | Vary (from 22.50–40 Gy to 159.2 Gy) | DUS | 2–13 years | Prevalence of carotid stenosis (CAS) <br> • >50% = 25% <br> • >70% = 12% <br> • carotid occlusion 4% <br> Cumulative incidence of carotid stenosis >50% <br> • 12-month = 4% <br> • 24-month = 12% <br> • 36-month = 21% |
| Carpenter et al. (2018) [32] | Retrospective cohort study (n = 366) | USA | 48±19Gy | DUS 95%, CTA 3%, MRA 1% | 4.1 years | Actuarial risk of Asymptomatic CAS> 50% at 8 years = 33.7% |

The data are presented as the mean ± standard deviation or as the number (percentage) unless otherwise stated. CAS, Carotid artery stenosis; Gy, Gray; DUS, duplex ultrasonography; CT, computer tomography; CTA, computer tomography angiography; MRA, Magnetic resonance angiography.

years was reported by Carpenter et al. (2018) [32] and Carpenter et al. (2023) [33]. Our report had a lower incidence than that of Texakalidis P et al. [31], which may have been the effect of using a different modality. The earlier cumulative incidence was reported in both of these studies. To find CAS that might have been discovered earlier than our investigation, Texakalidis P et al. [31] used doppler ultrasonography (DUS). There were, however, some differences, such as the radiation dose, which varied from a low dose (22.50–40 Gy) to a high dose (159.2 Gy) in Texakalidis P et al. [31] Even though the cumulative risk in the first 3 years of our study was lower than in the previously mentioned study, the carotid lesions progressed in 16.86% of cases and to significant stenosis in 3.7% during a mean follow-up of only 2.10 ± 1.36 years.

Tables 5 and 6 revealed that numerous studies were conducted in numerous countries, with a variety of ethnic groups, and that the radiation dosage, health access, and comorbidity profile varied. Studies comparing the ethnic groups in radiation-induced CAS have not been conducted but in atherosclerosis CAS, there was a study comparing the phenotypic differences between mainland Chinese and American Caucasian people with using magnetic resonance imaging (MRI) revealed that the Chinese had a significantly larger lipid/necrosis core and lesion characteristics such as diffuse intimal thickness or small eccentric plaque without calcification, while the American Caucasian people had a significantly higher amount of calcified plaque [34]. Therefore, there were some differences in plaque characteristics between races, which may have been caused by other factors like lifestyle choices or underlying diseases. Furthermore, comparing Tables 5 and 6, we discovered that the population in Sun L et al.'s study had more comorbidities than ours, including higher rates of smoking (83% versus 59%), hypertension (67% versus 23%), diabetes mellitus (22% versus 11%), and hyperlipidemia (51% versus 6%) [21]. As a result, the population's atherosclerotic risk was higher than ours. Although the radiation levels in the Asian population were similar, Makita C. et al.'s data from Japan showed higher ICVE than ours. However, there were some differences in comorbidities, such as higher smoking (85.6% versus 59%), higher mean age (69 versus 56 years), higher diabetes mellitus (25% versus 11%), higher hypertension (45% versus 22.8%), and higher dyslipidemia (33% versus 6%) [23]. Despite having the same level of radiation, our population has fewer comorbidities linked to atherosclerosis risk than other Asian populations, however, comparisons are still challenging because no study has comparable ethnic groups. However, different countries have varied radiation protocols, screening programs, and health access.

Therefore, the screening recommendations should consider the time elapsed since radiotherapy. However, many of the current guidelines, including the 2022 Society for Vascular Surgery (SVS) guideline, the 2017 American Society of Clinical Oncology clinical practice guideline for head and neck survivorship care, and the 2024 National Comprehensive Cancer Network guideline for HNC, do not specify the optimal timing and frequency of screening in this group [35–37] on the basis that the evidence does not appear to be sufficient to recommend routine screening for symptomatic patients who have previously received neck radiotherapy. Some guidelines address this issue. The 2021 International Cardio-Oncology Society guideline recommends that baseline risk factors, such as carotid calcification, be evaluated and optimized by routine CT imaging for staging or planning of radiotherapy [36]. Assessment by carotid DUS is recommended as early as one-year post-radiotherapy (determined in high-risk patients by cardiovascular risk), with follow-up ultrasonography every 3–5 years to guide preventive therapy [36]. The American Head and Neck Survivorship consensus statement recommends carotid DUS every 2–5 years [37]. Although the recommendation for screening remains debatable, the current evidence suggests that this population should be aware of CAS and its sequelae.

Although our population was mostly under the age of 65 years, 18% had pre-existing CAS and 3.63% had CAS >50%. The prevalence of CAS >50% in the general population has been

reported to be 2% [38], which is lower than in our population. However, a global meta-analysis from 2020 found that the prevalence of asymptomatic CAS >50% in patients aged 30–79 years was 1.5%, representing a 59% increase since 2000 [39], which may explain the higher frequency of CAS in our data. Before radiotherapy, 8.7% of our patients under the age of 65 years had CAS, which was significant (>50%) in 0.6% of cases. More than 50% of our study population were smokers and one-fifth had hypertension, with comorbidities including diabetes mellitus found in 10.9% of cases, hyperlipidemia in 6.28%, and obesity in 4.43%. Atherosclerosis was a significant risk factor for development and progression of carotid artery lesions. Although none of the published guidelines recommend routine screening, the European Society for Vascular Surgery (ESVS), European Society of Cardiology, SVS, and German–Austrian guidelines recommend screening in patients with multiple risk factors if they are identified to have significant stenosis and are being considered for carotid endarterectomy or carotid artery stenting [33,40–44]. The risk factors cited by the SVS include peripheral arterial disease, age >65 years, coronary artery disease, smoking, and hypercholesterolemia, whereas the other guidelines consider patients with no clinical evidence of atherosclerosis but at least two of hypertension, hyperlipidemia, smoking, family history of stroke, and early-onset atherosclerosis to be at increased risk of CAS [35,40–44]. The 2021 United States Preventive Services Task Force guidelines advise against any form of routine CAS screening but suggest that screening be considered in selected patients [45]. According to a recent meta-analysis, DUS has been the cornerstone of screening methods, and changes in intima-media thickness (IMT) are the primary marker of disease development. Additional screening methods included computer tomography angiography (CTA), magnetic resonance angiography (MRA), and total plaque score (TPS). It has been stated that arterial stiffness from DUS and serum biomarkers can be used; however, further research may be necessary [46]. Our study found a significant difference in mean age between patients with no carotid artery lesions (53.94 ± 13.56 years), those with a carotid artery lesion (69.64 ± 10.10 years), and those with significant (>50%) stenosis (74.09 ± 8.63 years). Age was an important risk factor for CAS, particularly age >65 years, when the risk of developing CAS after radiotherapy was increased by two-fold (aHR 2.60) in the group with no carotid lesions. However, Smith et al. found that the risk of ischemic stroke was higher in younger patients who underwent radiotherapy than in the general population [47]. Huang et al. also found that the risk of ischemic stroke was 1.8 times higher in patients aged younger than 55 years at the time of radiotherapy than in age-matched controls from the general population [48]. However, they found no significant difference in risk among those aged 55 years or older, possibly because of the higher competing risk of cancer and non-cancer-related mortality in older individuals. Another study demonstrated that the risk of ischemic stroke was age-dependent, with the highest incidence rate ratio in those aged younger than 40 years [49]. Chang et al. found that the plaque score was higher in younger patients (<41 years) than in older patients, indicating development of more severe atherosclerosis after radiotherapy in the younger age group [50]. Furthermore, we found that patients with diabetes mellitus and those with hypertension were at significantly increased risk of both CAS and significant (>50%) stenosis before radiotherapy. Furthermore, the risk of developing significant (>50%) CAS was significantly higher in patients with dyslipidemia, coronary artery disease, or peripheral arterial disease than in those without a carotid lesion. The major risk factors for radiation-induced CAS were outlined in a recent meta-analysis. These factors included radiation treatment, both dose and duration dependent, age over 50, time since radiation >9 years, cardiovascular risk factors (such as diabetes, hyperlipidemia, and smoking), positive human papillomavirus (HPV) status, and the presence of genotyping known as the "TC haplotype in rs662-rs705379 of single nucleotide polymorphisms (SNP) PON1" in NPC. [46]. Therefore, our data suggest that age >65 years are important risk factors for development of CAS in

patients with HNC who receive radiotherapy. Our literature search identified other reports suggesting that current smoking (hazard ratio [HR] 1.16), diabetes (HR 1.15), CAS (HR 22.18), and hypertension (HR 1.39) are associated with an increased risk of ischemic stroke in patients with HNC who receive radiotherapy [21,47]. Those reports also indicated that the presence of at least one uncontrolled cardiovascular risk factor at the time of diagnosis of HNC was associated with an increased risk (HR 1.09) of ischemic stroke following completion of radiotherapy.

The 2023 ESVS guideline stresses the importance of medical therapy and monitoring of risk factors in patients with asymptomatic CAS [40]. Their definition of "optimal medical therapy" includes lifestyle modification (diet, exercise, smoking cessation, and weight loss), control of hypertension, optimal glycemic control, low-dose aspirin (for prevention of late myocardial infarction and other cardiovascular events), and lipid-lowering therapy (for long-term prevention of stroke and other cardiovascular events), in particular statins [40]. In our study, use of antiplatelet agents, ACEIs, and statins was high in patients with pre-existing CAS and significant (>50%) stenosis. According to the 2023 ESVS guideline, antiplatelet therapy prevents myocardial infarction [40], whereas King et al. found that the benefit of antiplatelet therapy lies in its ability to reduce the risk of stroke (HR 0.45) [51]. According to a recent meta-analysis, blood pressure control, statins, and antiplatelet medication may be advantageous, although medical treatment has not been thoroughly studied [46]. Our patients who used antiplatelet agents had a significantly higher rate of pre-existing CAS and significant stenosis but no increased risk of developing CAS. Despite the controversy surrounding the use of antiplatelet agents in prevention, they may be a confounding factor because of comorbidities and preexisting CAS.

In patients with asymptomatic (50%–60%) CAS, it is reasonable to perform annual DUS for surveillance as recommended in the 2023 ESVS guideline [40]. The 2021 European Society of Cardiology guidelines recognize the coronary calcium score and carotid plaque/stenosis to be important risk factors [52]. Liu et al. proposed the use of the TPS for prediction of progression of CAS in patients with HNC and mild CAS after radiotherapy and found that a score ≥7 strongly predicted progression of CAS (OR 41.106) and a trend of imminent ischemic stroke that was higher than that in patients with a TPS <7 (p = 0.09) [53]. They recommended close monitoring during the first 2 years after radiotherapy in patients with HNC and a TPS ≥7 [53]. Our study found increases in TPS, wall thickness at all locations in the carotid artery (CCA, bulb, and ICA), and percent stenosis at several locations in this artery (proximal CCA, bulb, and ICA) at 1 month and 1 year after radiotherapy. Furthermore, the 2023 ESVS guidelines mention clinical/imaging criteria for identifying patients at higher risk of stroke in whom carotid endarterectomy or stenting may be considered, such as those with silent infarction on CT or magnetic resonance imaging, 20% progression, a large plaque area, plaque echolucency, and intraplaque hemorrhage [40].

More than half of the patient population in our study had TNM stage 4 pharyngeal cancer and received cisplatin chemotherapy. Patients with laryngeal cancer had a significantly greater number of preexisting CAS lesions and significant stenoses as well as a higher risk of developing CAS (aHR 2.36) in comparison with patients who did not have a carotid artery lesion. Furthermore, preexisting CAS was significantly less likely in patients with pharyngeal cancer and significantly more likely in those with cancer involving the salivary glands. Similarly, Liu et al. found that their patients without NPC had a 6-fold greater risk of developing significant CAS during follow-up, especially within 5 years after radiotherapy (12.7% vs 2.0%) [54]. Their study found that patients with laryngeal cancer had the lowest cumulative significant CAS-free rate of all cases of HNC. Despite adjustment for vascular risk factors, their group without NPC remained at higher risk of CAS. Cheng et al. found that patients with laryngeal cancer that did

not include NPC had a 6-fold higher risk of developing significant CAS than patients with other types of HNC [55], which they attributed to the better prognosis after frequent bilateral irradiation to the neck, meaning that patients survived for longer with increasing likelihood of carotid atherosclerosis. In their study, the group that had more cervical lymph node involvement required more aggressive radiotherapy. This result is in contrast with our finding that patients with laryngeal cancer were more likely to have preexisting CAS and progression of CAS whereas those with NPC did not have a significantly increased risk of developing CAS. Liu at al. found that their patients with NPC had a lower rate of smoking, which is a risk factor for atherosclerosis [54]. While our results are not influenced by staging, Tan et al. discovered that survivors with stage 1 NPC had a higher incidence of stroke [24]. There is considerable evidence indicating that cisplatin increases the risk of ischemic stroke [56] and plaque vulnerability [57]. A registry-based study of NPC survivors found that those treated with chemotherapy and radiotherapy had the highest risk of ischemic stroke (HR1.46) [58], although this has not been a consistent finding elsewhere [24] and our data.

Our study found that higher radiation doses did not increase the risk of developing CAS. The primary radiation technique used in our study was VMAT, which was administered at a mean total dose of 69.30 ± 24.29 Gy. Several other studies found no relationship between carotid artery dosimetry and ischemic stroke or CAS [23,32,33]. In contrast, Van Aken et al. found that the absolute ($cm^3$) V10 Gy–V50 Gy, relative (percent) V10 Gy–V30 Gy, and maximum radiation dose to the carotid arteries were associated with ICVE during a mean follow-up of 3.4 years [22]. Carpenter et al. found that the absolute V10 Gy (HR 1.09), V20 Gy (HR 1.1), V30 Gy (HR 1.1), V30 Gy (HR 1.1), V40 Gy (HR 1.09), V60 Gy (HR 1.11) and V70 Gy (HR 1.16) values were associated with asymptomatic CAS in multivariable analysis [33]. These findings are consistent with the suggestion that even a low dose of radiotherapy may increase the risk of asymptomatic CAS. [47] The use of the conformal radiotherapy technique with three-dimensional planning allows for at least partial sparing of the carotid artery. Radiotherapy for HNC have improved during the past decade and now includes photon beam therapy, such as intensity-modulated radiotherapy, the most advanced form of which is VMAT [53]. VMAT is inherently limited by the physical properties of the photon beam, resulting in unavoidable irradiation of normal tissues at low to moderate doses even at significant distances from the tumor [23,58–61]. However, our study did not find an association between radiation technique and the development of CAS.

In terms of location, atherosclerotic CAS is more likely to be found in the distal CCA near the bifurcation whereas radiotherapy-induced CAS can also involve the proximal CCA [47]. Moreover, stenotic lesions after radiotherapy tend to be longer, and the stenosis tends to be maximal at the end of the stenotic area [62]. These reports are consistent with our finding that radiation has an effect on the proximal CCA, bulb, and ICA, potentially leading to a longer lesion. Lam et al. found that the CCA and ICA were the vessels most commonly involved, followed by the ECA and vertebral artery [63]. Van Aken et al. reported that the carotid bulb and CCA showed the most significant dose-response in terms of development of ischemic stroke [22]. Furthermore, vessel damage after radiotherapy extended beyond the margin of the radiation field, often to the proximal CCA and distal ICA [63]. The presence of more proximal atherosclerosis has implications for surgical and endovascular management and can only be evaluated in a limited portion of the CCA [34,64]. Plaques exposed to radiation are more diffuse in appearance with less shadowing and are more hypoechoic [65]. It is presently believed that anechoic or hypoechoic plaque represents intraplaque hemorrhage or lipid deposits and an increased risk of stroke [66]. Fokkema et al. found that plaque post-radiotherapy showed less infiltration of macrophages and a smaller lipid core, which suggested that radiotherapy-induced plaque is more stable and less active than atherosclerotic lesions in the absence of

radiotherapy [67]. These findings are consistent with the fact that irradiated arteries are more prone to develop restenosis than non-irradiated vessels [68].

Radiation-induced CAS is characterized by damage to the three layers (intima, media, and adventitia) of the carotid artery, which causes inflammation and fibrosis, leading to intimal thickening [62]. Radiotherapy-induced CAS is mediated via multiple mechanisms, including direct vessel damage, accelerated atherosclerosis, intimal proliferation, necrosis of the media, and peri-adventitial fibrosis [47] which causes wall thickening and increases plaque score. Radiation causes the artery wall to become inflamed, which sets off an array of events involving endothelial cells, cytokines, and growth factors that alter the vascular wall. One of the most significant processes appears to be endothelial cell damage [69]. Endothelial dysfunction manifests before morphological changes [70]. The lack of endothelial nitric oxide synthase expression causes increased permeability, fibrin deposition in the extravascular space, and platelet adherence to the endothelium surface, all of which accelerate atherosclerosis [69,71]. The internal elastic lamina is then destroyed, and the endothelium noticeably thickens [69]. Platelets release basic fibroblast and platelet-derived growth factors, which encourage smooth cells to replicate in the media and migrate to the intima [69]. There, the smooth cells continue to proliferate, and an extracellular matrix is deposited, thickening the intima [69]. All of these alterations lead to changes in the arterial wall's structure, including luminal narrowing and changes in the vessel's compliance and distensibility [72]. Additionally, the level of inflammation has increased [73]. With the ability to eliminate Ox-LDL, monocytes can penetrate the vessel wall and develop into macrophages, which then enter the subendothelial region and produce foam cells [73]. A "fatty steak" is formed by the foam cells, T-lymphocytes, and smooth muscle cells. Fibrous plaque is the result of subsequent matrix synthesis [73]. Tumor necrosis factor α, interferon γ, transforming growth factor β, and nuclear factor kappa B activation are additional variables involved [73]. As a late consequence of radiation, CAS may also be caused by oxidative stress and inflammation [74]. When pre-existing atherosclerosis was exposed to radiation, the consequence was smaller, macrophage-rich plaques with increased apoptosis and intraplaque bleeding [75]. Radiation can damage the adventitia's vasa vasarum, which diminishes blood flow and induces ischemia necrosis [76]. As a result, muscle fibers and elastic tissue are lost, and fibrosis takes their place [77]. Additionally, extrinsic compression is seen due to a notable thickening of the endothelium and adjacent periadventitial fibrosis [77]. The deposition of fibrin in the medial and intimal layers, as well as the progressive replacement of that fibrin by collagen as a result of increased artery exposure to ionizing radiation, combine to produce intima-media thickness (IMT) [78]. All of these conditions result in arterial wall thickening, arterial stiffness, stenosis, formation of plaque, thrombosis, and occlusion or disruption of blood flow [69].

Numerous biomarker types have been proposed for use in radiation-induced CAS, including imaging biomarkers (IMT, measuring the beta-stiffness index (B) and elastic modulus from DUS) and serum biomarkers (Interferon-6, Interferon-1b, Tumor necrosis factor-α, Ox-LDL, and adipokines) as well as additional techniques like speckle ultrasound, contrast-enhanced ultrasound, and positron emission tomography-computed tomography [79]. Although this biomarker was not measured in our investigation, wall thickness was an imaging result that was indirectly related to IMT and may be utilized with total plaque score to predict radiation-induced CAS from CT scans. Many guidelines recommend DUS screening that might be pitfall due to the extensive effect to proximal CCA and distal ICA. According to certain research, screening for CAS may be cost-effective if the prevalence in the specific group is 20% or above [80]. Two meta-analyses showed that the prevalence of CAS >50% in HNC patients treated with RT is probably more than 20%, proving the cost-effectiveness of screening [29,31]. The surveillance frequency interval is not well defined. Screening should ideally

start before patients are at risk of experiencing symptoms of CAS [46]. Carotid DUS has shown changes in the IMT and carotid artery lumen as early as six months and a year, respectively [81,82]. Radiation-induced CAS may start as early as one year after treatment, even though CAS and ICVE have always been considered to be sequelae more than six years after radiation [31]. A study suggested CT-assisted ultrasound assessment to improve assessment accuracy because of increased intra-plaque calcification in radiation-induced CAS [83], as previously mentioned, due to the broad influence on proximal CCA and distal ICA. According to our study, the cumulative incidence of ICVE and CAS >50% in Thailand is modest. The cost-effectiveness of routine screening should be discussed, but clinicians should be concerned if there is evidence of CAS in imaging over the tumor follow-up period. In a setting with limited resources, screening should concentrate on patients with high-risk factors and a history of carotid artery lesions before the start of radiation therapy. DUS should be used one year after the end of radiation therapy, and the plaque characteristics should be taken into consideration when combining CT scans for routine tumor follow-up.

This study had several limitations. First, many of the patients were followed up at other hospitals, which meant that our long-term data were incomplete for this group as the missing data. This resulted in information bias, which we addressed by utilizing right-censored data and analyzing it using the Cox regression model. Second, clinical decision-making may have been influenced by other risk factors for vascular disease resulting in selection bias. Furthermore, the use of and compliance with prescribed medications may have been confounding factors. Third, the study had a retrospective cohort study design, and the data analyzed were extracted from medical records. Therefore, unrecognized confounding factors could have introduced bias, which would weaken our conclusions. However, to identify confounding factors, we use multivariate Cox regression to examine the interaction with the atherosclerotic risk factors. A prospective study with a standardized protocol is needed for more accurate identification of risk factors for CAS in patients with HNC who undergo radiotherapy. Fourth, we evaluated vascular status on contrast-enhanced CT or magnetic resonance scans that were obtained using various protocols for tumor follow-up, which could also have been a source of bias. Due to the limited resolution, small plaques might have been overlooked. Finally, we analyzed the study data retrospectively, which resulted in variable clinician decision-making concerning the radiotherapy protocol and not all relevant data were available, particularly for patients who underwent long-term follow-up. However, the identification of the pathogenesis via which patients with laryngeal cancer become vulnerable to development of CAS was not the aim of this study. Large multicenter prospective studies are needed to clarify the risk factors for radiotherapy-induced CAS in patients with HNC and to determine an appropriate screening protocol.

## Conclusions

This study found that the incidence of ICVE and the cumulative incidence of CAS were lower in the Thai population with HNC than in other populations. The main risk factors for new CAS were age >65 years, laryngeal cancer, and total plaque score. Changes in the carotid artery were detected early and could involve any location in the vessel. Clinicians should focus on vascular surveillance and monitoring during follow-up of patients with these vascular risk factors after completion of radiotherapy for HNC.

## Author Contributions

**Conceptualization:** Nawaphan Taengsakul, Padungcharn Nivatpumin, Thong Chotchutipan, Sunanta Tungfung.

**Data curation:** Nawaphan Taengsakul, Padungcharn Nivatpumin, Thong Chotchutipan, Sunanta Tungfung.

**Formal analysis:** Nawaphan Taengsakul.

**Funding acquisition:** Nawaphan Taengsakul, Sunanta Tungfung.

**Investigation:** Nawaphan Taengsakul, Sunanta Tungfung.

**Methodology:** Nawaphan Taengsakul, Padungcharn Nivatpumin.

**Project administration:** Nawaphan Taengsakul.

**Resources:** Thong Chotchutipan.

**Supervision:** Padungcharn Nivatpumin, Thong Chotchutipan, Sunanta Tungfung.

**Validation:** Nawaphan Taengsakul, Padungcharn Nivatpumin, Thong Chotchutipan, Sunanta Tungfung.

**Visualization:** Nawaphan Taengsakul, Padungcharn Nivatpumin, Thong Chotchutipan, Sunanta Tungfung.

**Writing – original draft:** Nawaphan Taengsakul, Thong Chotchutipan, Sunanta Tungfung.

**Writing – review & editing:** Nawaphan Taengsakul.

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
