## [Decision Letter · Decision Letter 0]

8 Oct 2024

PONE-D-24-33754Carotid artery stenosis and ischemic cerebrovascular events after radiotherapy in patients with head and neck cancerPLOS ONE

Dear Dr. Taengsakul,

Thank you for submitting your manuscript to PLOS ONE. After careful consideration, we feel that it has merit but does not fully meet PLOS ONE’s publication criteria as it currently stands. Therefore, we invite you to submit a revised version of the manuscript that addresses the points raised during the review process.

The authors should carefully address, point by point, all the points raised by the reviewers, in order for the manuscript to be considered for publication. 

We look forward to receiving your revised manuscript.

Kind regards,

Paula Boaventura, PhD

Academic Editor

PLOS ONE

Journal requirements: When submitting your revision, we need you to address these additional requirements. 1. Please ensure that your manuscript meets PLOS ONE's style requirements, including those for file naming. The PLOS ONE style templates can be found at https://journals.plos.org/plosone/s/file?id=wjVg/PLOSOne_formatting_sample_main_body.pdf and https://journals.plos.org/plosone/s/file?id=ba62/PLOSOne_formatting_sample_title_authors_affiliations.pdf 2. As are reporting a retrospective study of medical records or archived samples, please ensure that you have discussed whether all data were fully anonymized before you accessed them and/or whether the IRB or ethics committee waived the requirement for informed consent. If patients provided informed written consent to have data from their medical records used in research, please include this information. 3. We note that your Data Availability Statement is currently as follows: [All relevant data are within the manuscript and its Supporting Information files.] Please confirm at this time whether or not your submission contains all raw data required to replicate the results of your study. Authors must share the “minimal data set” for their submission. PLOS defines the minimal data set to consist of the data required to replicate all study findings reported in the article, as well as related metadata and methods (https://journals.plos.org/plosone/s/data-availability#loc-minimal-data-set-definition). For example, authors should submit the following data: - The values behind the means, standard deviations and other measures reported;- The values used to build graphs;- The points extracted from images for analysis. Authors do not need to submit their entire data set if only a portion of the data was used in the reported study. If your submission does not contain these data, please either upload them as Supporting Information files or deposit them to a stable, public repository and provide us with the relevant URLs, DOIs, or accession numbers. For a list of recommended repositories, please see https://journals.plos.org/plosone/s/recommended-repositories. If there are ethical or legal restrictions on sharing a de-identified data set, please explain them in detail (e.g., data contain potentially sensitive information, data are owned by a third-party organization, etc.) and who has imposed them (e.g., an ethics committee). Please also provide contact information for a data access committee, ethics committee, or other institutional body to which data requests may be sent. If data are owned by a third party, please indicate how others may request data access.

Reviewers' comments:

Reviewer's Responses to Questions

**Comments to the Author**

1. Is the manuscript technically sound, and do the data support the conclusions?

Reviewer #1: Yes

Reviewer #2: Partly

2. Has the statistical analysis been performed appropriately and rigorously? 

Reviewer #1: Yes

Reviewer #2: No

3. Have the authors made all data underlying the findings in their manuscript fully available?

Reviewer #1: Yes

Reviewer #2: Yes

4. Is the manuscript presented in an intelligible fashion and written in standard English?

Reviewer #1: Yes

Reviewer #2: Yes

5. Review Comments to the Author

Reviewer #1: According to the authors, “the purpose of this cross-sectional study was to determine the incidence of ICVE and carotid artery stenosis (CAS) in patients with HNC who receive radiotherapy and the risk factors for CAS”. To this end, they enrolled 907 patients with HNC who underwent radiotherapy between February 2011 and June 2022. It is of particular note that all patients enrolled were from Thailand which may be significant in regard to their conclusions. This manuscript is well written and the results and conclusions are well documented.

Major Concerns:

There are many other publications investigating the connection between patients treated with radiation therapy for head and neck cancers and the incidence of CAS and ICVE. Many of the papers have differing conclusions and there seem to be a lot of studies that find differences only based on the population under study. The current manuscript appears to be one of those. They found significant differences in the Thai population, but recent publications by Samara 2023 (https://ar.iiarjournals.org/content/43/12/5657.long) and Rosen 2024 (https://www.sciencedirect.com/science/article/pii/S1368837524001945?via%3Dihub) found contradictory results. Furthermore, neither of these publications is referenced in the current manuscript. The author’s main conclusion is that patients having undergone radiation therapy for HNC should be monitored in follow-up visits, which is a true and valid finding and many of the previous publications also expressed this same conclusion.

The novelty and applicability of this study to clinical research is not as intense with so many similar publications in the literature.

Minor Concerns:

1. Table 2 lacks P values for females and for ages ≤65.

2. Table 3 lacks P values for ages ≤65.

3. Line 299, reference should be US Preventative Services Task Force, JAMA 2021.

4. Line 375, reference 53 and 43 are the same, Chu et al. Br J Cancer 2011.

5. References after 52 will need to be adjected.

6. Reference 68, Fokkema 2012, is not cited in manuscript.

Reviewer #2: Abstract

1. The abstract mentions relative risks (RR) for various factors associated with carotid artery stenosis (CAS) (e.g., RR of 2.32 for laryngeal cancer and RR of 2.26 for age >65 years), but it does not provide 95% confidence intervals and p values.

Study Design and Methodology

1. The manuscript describes that 907 patients were enrolled. What was the strategy for addressing missing data from patients who could not complete follow-up? How might this impact the reliability of the results?

2. The study has a cross-sectional design, but it follows patients longitudinally through to the end of 2023. Is there a rationale for not utilizing a cohort or case-control design, which may have offered more robust data on causality?

3. The authors note that laboratory data was incomplete, limiting statistical analysis. How significant are these missing data for the study's conclusions, especially in understanding the risk factors? Could imputation methods have been used to address this?

Statistical Analysis

1. The study uses a generalized linear model to calculate risk ratios, but the rationale for this choice is not discussed. Why was this model chosen over alternative models such as Cox proportional hazards, especially given that survival analysis is also performed? Can the authors provide further justification or compare the results with alternative modeling techniques?

2. Although some potential confounders, such as age and comorbidities, are addressed, the manuscript does not clearly indicate how interactions between variables were managed. Were interactions between key risk factors like age, cancer type, and diabetes mellitus evaluated?

3. The follow-up duration varies, and events such as carotid artery stenosis (CAS) and ischemic cerebrovascular events (ICVE) are time-dependent. Would the results be affected if sensitivity analyses were performed to control for different follow-up times across patients? Was a time-to-event analysis considered?

Results and Discussion

1. The manuscript references several studies showing a higher incidence of CAS and ICVE post-radiotherapy in head and neck cancer patients. How do the authors reconcile the discrepancy between their lower reported rates and those in previous studies? Could differences in radiation dosages or imaging modalities used for detection account for this?

2. The study finds a significant increase in wall thickness and plaque scores post-radiotherapy. Is there a biological explanation for these findings beyond what is already known? Could specific biomarkers of radiation-induced vascular injury have been explored to add depth to this analysis?

3. The study focuses on a Thai population. How applicable are the results to other ethnic groups, especially in terms of radiation dosages, healthcare access, and comorbidity profiles? Can the authors discuss this limitation more thoroughly?

4. The manuscript mentions potential mechanisms of radiation-induced vascular damage but could benefit from a deeper discussion of current advancements in radiobiology. For example, what role might inflammation or endothelial dysfunction play in exacerbating CAS after radiotherapy?

5. The study suggests vascular surveillance, but there is no mention of the cost-effectiveness or practicality of implementing routine surveillance in clinical practice. How feasible would it be to integrate these recommendations into standard care for head and neck cancer patients, particularly in resource-limited settings?

6. The discussion section outlines some limitations, such as incomplete follow-up for some patients and lack of standardization in CT protocols. However, were there any biases introduced by the retrospective nature of the study, especially in clinical decision-making during follow-up care?

6. PLOS authors have the option to publish the peer review history of their article (what does this mean?). If published, this will include your full peer review and any attached files.

Reviewer #1: No

Reviewer #2: No

---

## [Author Response · Author response to Decision Letter 0]

23 Oct 2024

Dear Editor

 Thank you very much for the excellent comments and suggestions from the reviewers. We have revised our manuscript according to their suggestions as shown below.

Reviewer 1: 

Major concern

1. There are many other publications investigating the connection between patients treated with radiation therapy for head and neck cancers and the incidence of CAS and ICVE. Many of the papers have differing conclusions and there seem to be a lot of studies that find differences only based on the population under study. The current manuscript appears to be one of those. They found significant differences in the Thai population, but recent publications by Samara 2023 (https://ar.iiarjournals.org/content/43/12/5657.long) and Rosen 2024 (https://www.sciencedirect.com/science/article/pii/S1368837524001945?via%3Dihub) found contradictory results. Furthermore, neither of these publications is referenced in the current manuscript. The author’s main conclusion is that patients having undergone radiation therapy for HNC should be monitored in follow-up visits, which is a true and valid finding and many of the previous publications also expressed this same conclusion.

The novelty and applicability of this study to clinical research is not as intense with so many similar publications in the literature.

Response: We appreciate your recommendation, and we have implemented the most recent reference you suggested at Line 265-266, 343-348, 367-372,389-391 with References 30 and 46.

Minor concern

1. Table 2 lacks P values for females and for ages ≤65.

Response: We used chi-square to compare the risk ratios between age groups and sexes, starting with those with lower proportions (females and those over 65), so that neither variable had a p-value because it was the baseline value.

2. Table 3 lacks P values for ages ≤65.

Response: By using chi-square to compare the risk ratios of age groups, we were able to exclude p-values for variables that had baseline values

3. Line 299, reference should be US Preventative Services Task Force, JAMA 2021.

Response: We changed the reference to the correct one as mentioned.

4. Line 375, reference 53 and 43 are the same, Chu et al. Br J Cancer 2011.

Response: We modified the latter reference and fixed the reference that was repeated.

5. References after 52 will need to be adjected.

Response: After reference number 52, we modified the reference.

6. Reference 68, Fokkema 2012, is not cited in manuscript.

Response: In our initial version, we had misdefined this reference, which we have now amended and referenced to the proper sentence.

Reviewer 2: 

Abstract

1. The abstract mentions relative risks (RR) for various factors associated with carotid artery stenosis (CAS) (e.g., RR of 2.32 for laryngeal cancer and RR of 2.26 for age >65 years), but it does not provide 95% confidence intervals and p values.

Response: We switched the statistical analysis to the Cox proportional hazard model as described at lines 131-135, which altered the relative risk to the adjusted hazard ratio at lines 36–49,42,187-189, 270-271, 392,427-429, 433, 435-443, 553 and table 3 as well as the influence risk factor and all statistical values.

Study Design and Methodology

1. The manuscript describes that 907 patients were enrolled. What was the strategy for addressing missing data from patients who could not complete follow-up? How might this impact the reliability of the results?

Response: We changed the statistical analysis from a generalized linear model to Cox proportion hazard ratio with right-censored data to improve reliability from these missing data due to incomplete follow-up and we defined the number and percentage of missing data at lines 73-75.

2. The study has a cross-sectional design, but it follows patients longitudinally through to the end of 2023. Is there a rationale for not utilizing a cohort or case-control design, which may have offered more robust data on causality?

Response: We shifted to a retrospective cohort study at lines 72 and 534 because the initial manuscript's identification was incorrect. 

3. The authors note that laboratory data was incomplete, limiting statistical analysis. How significant are these missing data for the study's conclusions, especially in understanding the risk factors? Could imputation methods have been used to address this?

Response: Only 14 of the cases in our data had complete laboratory data, therefore we were unable to analyze the data owing to significant data loss and inappropriate use of the imputation approach. However, we focus on analysis derived from underlying disease diagnosis, which is more accurate than test results.

Statistical Analysis

1. The study uses a generalized linear model to calculate risk ratios, but the rationale for this choice is not discussed. Why was this model chosen over alternative models such as Cox proportional hazards, especially given that survival analysis is also performed? Can the authors provide further justification or compare the results with alternative modeling techniques?

Response: As stated in lines 130-132, we modified the use of right-censored data with survival analysis using the Cox proportion hazard model and corrected data in Table 3.

2. Although some potential confounders, such as age and comorbidities, are addressed, the manuscript does not clearly indicate how interactions between variables were managed. Were interactions between key risk factors like age, cancer type, and diabetes mellitus evaluated?

Response: Our analysis of risk factor interactions to find confounding factors revealed that age, diabetes mellitus, and cancer type did not interact, as indicated in lines 132-134. 

3. The follow-up duration varies, and events such as carotid artery stenosis (CAS) and ischemic cerebrovascular events (ICVE) are time-dependent. Would the results be affected if sensitivity analyses were performed to control for different follow-up times across patients? Was a time-to-event analysis considered?

Response: We adjusted the influence from various follow-up times and defined time-to-event analysis using the right-censored data and the Cox proportional hazard ratio.

Results and Discussion

1. The manuscript references several studies showing a higher incidence of CAS and ICVE post-radiotherapy in head and neck cancer patients. How do the authors reconcile the discrepancy between their lower reported rates and those in previous studies? Could differences in radiation dosages or imaging modalities used for detection account for this?

Response: Large retrospectives in the USA showed a greater incidence of prevalence for only stroke, however, this study omitted information on follow-up duration and radiation dosage Two publications, including Makita C. et al. and Van Aken et al., had a similar level of percentage that involved stroke and TIA. We discovered that Van Aken et al. had a lower radiation dose, which may have contributed to their slightly lower incidence of ICVE than Makita C. et al. publication. Tan TH. et al.'s prior study with a 10-year incidence of stroke and no definition of radiation dose had the lowest incidence of ICVE. Despite a higher radiation exposure, as seen in Table 5, our investigation revealed the lowest 5-year cumulative incidence of both stroke and TIA when compared to the prior study.

In part, carotid artery stenosis was reported to have a prevalence of CAS (>50%) at a comparable rate. We discovered that Carpenter et al. (2018) and Carpenter et al. (2023) reported the cumulative incidence for more extended follow-up as 5 years, 8 years, and 10 years. We discovered that our report had a lower incidence than that of Texakalidis P et al., which may have been the effect of using a different modality. The earlier cumulative incidence was reported in both of these studies. To find CAS that might have been discovered earlier than our investigation, Texakalidis P et al. used ultrasonography. There were, however, some differences, such as the radiation dose, which varied from a low dose (22.50–40 Gy) to a high dose (159.2 Gy) in Texakalidis P et al.

In our manuscript, we have included a discussion of those issues at Lines 238-247, 270-278 along with Tables 5 and 6.

2. The study finds a significant increase in wall thickness and plaque scores post-radiotherapy. Is there a biological explanation for these findings beyond what is already known? Could specific biomarkers of radiation-induced vascular injury have been explored to add depth to this analysis?

Response: We provide a brief overview of the biomarker at Lines 502-508 and describe the pathophysiology of radiation-induced carotid artery stenosis. Wall thickness was an imaging result that was indirectly associated with intima-media thickness, even though this biomarker was not evaluated in our study. It may be used to predict radiation-induced carotid artery stenosis from CT scans.

3. The study focuses on the Thai population. How applicable are the results to other ethnic groups, especially in terms of radiation dosages, healthcare access, and comorbidity profiles? Can the authors discuss this limitation more thoroughly?

Response: We included a discussion of other ethnic groups, comparing our study to theirs in terms of comorbidity profile, radiation exposure, and health access at lines 287-308. This included the care to lessen their impact, as well as the limitations from several problems, like the loss of the screening routine for radiation-induced carotid artery stenosis at lines 529-540.

4. The manuscript mentions potential mechanisms of radiation-induced vascular damage but could benefit from a deeper discussion of current advancements in radiobiology. For example, what role might inflammation or endothelial dysfunction play in exacerbating CAS after radiotherapy?

Response: At Lines 473-501, we included this information regarding the radiobiology and mechanisms that contribute to the exacerbation of CAS following radiotherapy.

5. The study suggests vascular surveillance, but there is no mention of the cost-effectiveness or practicality of implementing routine surveillance in clinical practice. How feasible would it be to integrate these recommendations into standard care for head and neck cancer patients, particularly in resource-limited settings?

Response: In lines 510–527, we offer a suggestion for the limited resources situation and discuss the cost-effectiveness of surveillance.

6. The discussion section outlines some limitations, such as incomplete follow-up for some patients and a lack of standardization in CT protocols. However, were there any biases introduced by the retrospective nature of the study, especially in clinical decision-making during follow-up care?

Response: We added the discussion of biases brought about by the study's retrospective design in lines 536–542, particularly concerning clinical decision-making during follow-up care and limitations that we neglected to address in the original publication.

Thank you again for your kind consideration of our manuscript and very much look forward to hearing from you soon.

Sincerely,

Nawaphan Taengsakul

Department of Surgery, Chulabhorn Hospital, Chulabhorn Hospital, Princess Srisavangavadhana College of Medicine, Chulabhorn Royal Academy, 906 Kamphaeng Phet 6 Rd., Talat Bang khen, Lak Si, Bangkok 10210, Thailand. 

Tel +66-2-576-6791

Fax +66-2-576-6791

Email nawaphan.tan@cra.ac.th

---

## [Decision Letter · Decision Letter 1]

19 Nov 2024

Carotid artery stenosis and ischemic cerebrovascular events after radiotherapy in patients with head and neck cancer

PONE-D-24-33754R1

Dear Dr. Taengsakul,

We’re pleased to inform you that your manuscript has been judged scientifically suitable for publication and will be formally accepted for publication once it meets all outstanding technical requirements.

Kind regards,

Paula Boaventura, PhD

Academic Editor

PLOS ONE

Additional Editor Comments (optional):

Reviewers' comments:

Reviewer's Responses to Questions

**Comments to the Author**

1. If the authors have adequately addressed your comments raised in a previous round of review and you feel that this manuscript is now acceptable for publication, you may indicate that here to bypass the “Comments to the Author” section, enter your conflict of interest statement in the “Confidential to Editor” section, and submit your "Accept" recommendation.

Reviewer #1: All comments have been addressed

Reviewer #2: All comments have been addressed

2. Is the manuscript technically sound, and do the data support the conclusions?

Reviewer #1: Yes

Reviewer #2: Yes

3. Has the statistical analysis been performed appropriately and rigorously? 

Reviewer #1: Yes

Reviewer #2: Yes

4. Have the authors made all data underlying the findings in their manuscript fully available?

Reviewer #1: Yes

Reviewer #2: Yes

5. Is the manuscript presented in an intelligible fashion and written in standard English?

Reviewer #1: Yes

Reviewer #2: Yes

6. Review Comments to the Author

Reviewer #1: (No Response)

Reviewer #2: I have reviewed all the answers. All comments have been addressed. It seems that no other revisions need to be made.

7. PLOS authors have the option to publish the peer review history of their article (what does this mean?). If published, this will include your full peer review and any attached files.

Reviewer #1: No

Reviewer #2: No

---

## [Editor Report · Acceptance letter]

25 Nov 2024

PONE-D-24-33754R1 

PLOS ONE

Dear Dr. Taengsakul, 

I'm pleased to inform you that your manuscript has been deemed suitable for publication in PLOS ONE. Congratulations! Your manuscript is now being handed over to our production team.

Kind regards, 

on behalf of

Dr. Paula Boaventura 

Academic Editor

PLOS ONE